# PROBABILISTIC MODEL-BASED DYNAMIC ARCHITECTURE SEARCH

## ABSTRACT

The architecture search methods for convolutional neural networks (CNNs) have shown promising results. These methods require significant computational resources, as they repeat the neural network training many times to evaluate and search the architectures. Developing the computationally efficient architecture search method is an important research topic. In this paper, we assume that the structure parameters of CNNs are categorical variables, such as types and connectivities of layers, and they are regarded as the learnable parameters. Introducing the multivariate categorical distribution as the underlying distribution for the structure parameters, we formulate a differentiable loss for the training task, where the training of the weights and the optimization of the parameters of the distribution for the structure parameters are coupled. They are trained using the stochastic gradient descent, leading to the optimization of the structure parameters within a single training. We apply the proposed method to search the architecture for two computer vision tasks: image classification and inpainting. The experimental results show that the proposed architecture search method is fast and can achieve comparable performance to the existing methods.

## 1 INTRODUCTION

Convolutional neural networks (CNNs) make remarkable progress in various computer vision tasks. As researchers have developed deeper architectures and new components to improve performance, the architecture of CNNs is becoming complicated. Since numerous modules exist to construct CNN architectures, designing the appropriate CNN architecture for a target problem is critical. However, finding a better combination of basic modules for CNNs, e.g., convolutional and pooling layers, requires tremendous trial and error.

In this context, the methods for designing deep neural network architectures are actively proposed. Most existing methods treat the structure parameters, such as the type of layer and the connectivity, as hyper-parameters and optimize them by reinforcement learning (Zoph & Le, 2017) or evolutionary algorithms (Real et al., 2017; Suganuma et al., 2017). These methods search for a better architecture that maximizes the performance for validation data in the hyper-parameter optimization manner, i.e., they need neural network training for an architecture evaluation. This approach has succeeded in finding the state-of-the-art CNN architectures; however, it is computationally inefficient in general. Reducing the computational cost of the architecture search is critical for practical usage. Several pieces of research have focused on reducing the computational cost of architecture search (Liu et al., 2018b; Pham et al., 2018) by reusing the trained weights on different architectures and simultaneously optimizing the weights and the structure parameters. Another practical requirement is that the architecture search methods have fewer hyper-parameters to reduce the total effort of the architecture search process.

This work aims to develop an efficient architecture search method with fewer hyper-parameters. As neural network architectures can be represented by a sequence of categorical variables (Pham et al., 2018; Suganuma et al., 2017; 2018), we consider the neural networks consist of categorical variables of the structure parameters and continuous variables of the weights. Introducing multivariate categorical distribution as the underlying distribution for the structure parameters, we formulate an expected loss function under the distribution that is differentiable with respect to (w.r.t.) both the weights and the parameters of the distribution. We iteratively update the weights and the param-

eters of the distribution to the gradient steps within a single training, realizing a computationally efficient architecture search. The idea of the proposed method is based on Shirakawa et al. (2018), which instantiates the algorithm using the multivariate Bernoulli distribution and has proofed the concept on the simple neural network structure optimization. We extend their work and make it possible to search flexible architectures. We call the algorithm proposed in this paper the probabilistic model-based dynamic architecture search (PDAS). The straightforward probabilistic modeling of the architectures adopted in this work leads to the simple update rule of the parameters of the distribution, which relates to the stochastic natural gradient method (Ollivier et al., 2017). Moreover, we establish the parameter-adaptive architecture search method by injecting the learning rate adaptation mechanism of the stochastic natural gradient proposed in Nishida et al. (2018).

We apply the PDAS to search the architectures for two computer vision tasks: image classification and inpainting. The experimental results show that the PDAS is fast and can achieve comparable performance to the existing architecture search methods on both tasks.

The contribution of this paper is as follows: (1) we derive the algorithm for categorical distribution and make it possible to apply the framework proposed in Shirakawa et al. (2018) to the architecture search spaces represented by categorical variables. To the best of our knowledge, the natural gradient of categorical distribution has not been introduced in the context of stochastic natural gradient methods. (2) we show that PDAS, which has fewer hyper-parameters than efficient neural architecture search (ENAS) (Pham et al., 2018), is fast and can reach state-of-the-art performance. The intrinsic hyper-parameters of PDAS are the sample size and the learning rate, but the learning rate can be adaptive.

## 2 Probabilistic Model-Based Dynamic Architecture Search

**Formulation:** Following Shirakawa et al. (2018), we consider the neural network $\phi(W, M)$ having two different sets of parameters: the connection weights $W \in \mathcal{W}$ and the structure parameters $M \in \mathcal{M}$. We assume that the weights are real-valued and differentiable for loss function; however, the structure parameters can be discrete, i.e., the loss function is not differentiable w.r.t. these parameters. Our original objective is minimizing the loss $\mathcal{L}(W, M) = \int_{\mathcal{D}} l(z, W, M) p(z) \mathrm{d}z$, where $\mathcal{D}$ and $l(z, W, M)$ indicate the dataset and the loss function value of a datum $z$, respectively. Introducing a family of probability distributions $p_\theta(M)$ of $M$ parametrized by a real-valued vector $\theta \in \Theta$, we formulate a minimization of the expected loss under $p_\theta(M)$, namely

$$\mathcal{G}(W, \theta) = \int_{\mathcal{M}} \mathcal{L}(W, M) p_\theta(M) \mathrm{d}M \ , \tag{1}$$

where $\mathrm{d}M$ is a reference measure on $\mathcal{M}$. That is, we try to minimize the loss $\mathcal{L}(W, M)$ indirectly by minimizing $\mathcal{G}(W, \theta)$. This formulation is inspired from the recently-introduced black-box optimization framework called *Information Geometric Optimization* (Ollivier et al., 2017). The point is, one can choose the family of probability distributions so that $\mathcal{G}$ is differentiable w.r.t. both $W$ and $\theta$. It allows us to employ a gradient descent to optimize $W$ and $\theta$ simultaneously within a single training process.

We apply a stochastic gradient descent to minimize $\mathcal{G}$ on $\mathcal{W} \times \Theta$ equipped with the Fisher metric on $\Theta$. That is, we take the *vanilla* gradient w.r.t. $W$ and the *natural* gradient (Amari, 1998) w.r.t. $\theta$. They can be approximated by Monte-Carlo using $\lambda$ samples drawn from $p_\theta(M)$ and mini-batch loss $\mathcal{L}(W, M) \approx \bar{\mathcal{L}}(W, M) = \bar{N}^{-1} \sum_k^{\bar{N}} l(z_k, W, M)$ with mini-batch size $\bar{N}$. Then, we get

$$\nabla_W \mathcal{G}(W, \theta) \approx \frac{1}{\lambda} \sum_{n=1}^{\lambda} \nabla_W \bar{\mathcal{L}}(W, M_n) \ , \tag{2}$$

$$\tilde{\nabla}_\theta \mathcal{G}(W, \theta) \approx \frac{1}{\lambda} \sum_{n=1}^{\lambda} \bar{\mathcal{L}}(W, M_n) \tilde{\nabla}_\theta \ln p_\theta(M_n) \ , \tag{3}$$

where $\tilde{\nabla}_\theta \ln p_\theta(M) = F^{-1}(\theta) \nabla_\theta \ln p_\theta(M)$ is the natural gradient of the log-likelihood $\ln p_\theta(M)$, and $F(\theta)$ is the Fisher information matrix of $p_\theta(M)$. Shirakawa et al. (2018) proposed the framework that simultaneously optimizes both $W$ and $\theta$ by using approximated gradients of (2) and (3), and they instantiated the algorithm using the multivariate Bernoulli distribution. In other words,

they used the binary vector to select the network components, such as the unit, layer, and type of activation.

**Categorical Parameters:** In this paper, we consider categorical parameters as they can represent more flexible network architecture than binary parameters. We extend the previous work by introducing the multivariate categorical distribution as $p_\theta(M)$. We denote the $D$-dimensional categorical variables by $h = (h_1, \ldots, h_D)^{\mathrm{T}}$ and the number of categories for the $i$-th variable by $K_i (> 1)$. Let us introduce the one-hot representation of $h_i$, denoted as $m_i \in \{0, 1\}^{K_i}$, where the entries of $m_i$ are all zero but for the $h_i$-th entry, which is one. Our structure parameter vector $M$ is then written as $M = (m_1, \ldots, m_D)^{\mathrm{T}}$.

We introduce the multivariate categorical distribution $p_\theta$ as the underlying distribution for $M$, whose probability mass is $p_\theta(M) = \prod_{i=1}^{D} \prod_{j=1}^{K_i} (\theta_{ij})^{m_{ij}}$, where $\theta_{ij} \in [0, 1]$ represents the probability of $m_{ij}$ to be 1 and must satisfy $\sum_{j=1}^{K_i} \theta_{ij} = 1$. The natural gradient of the log-likelihood for the above distribution is given by $\tilde{\nabla}_\theta \ln p_\theta(M) = M - \theta$, since the natural gradient of the log-likelihood for an exponential family with sufficient statistics $T(M)$ with the expectation parameterization $\theta = \mathbb{E}[T(M)]$ is known to be $\tilde{\nabla}_\theta \ln p_\theta(M) = T(M) - \theta$ and the above parametrized distribution is an exponential family with the expectation parameterization. See Appendix A for details.

With this natural gradient of log-likelihood, we get the update rule of $\theta$ as follows:

$$\theta^{(t+1)} = \theta^{(t)} + \frac{\eta_\theta}{\lambda} \sum_{n=1}^{\lambda} u_n (M_n - \theta^{(t)}) \ , \tag{4}$$

where $\eta_\theta$ is the learning rate. As done in Ollivier et al. (2017); Shirakawa et al. (2018), we transform the loss value $\bar{\mathcal{L}}(W, M_n)$ into the utility $u_n$ to make the update invariant for the scale of the objective values. We use the following ranking-based utility transformation: $u_n = 1$ for best $\lceil \lambda/4 \rceil$ samples, $u_n = -1$ for worst $\lceil \lambda/4 \rceil$ samples, and $u_n = 0$ otherwise. The update rule (4) is the generalization of the case for Bernoulli distribution and ensures $\sum_{j=1}^{K_i} \theta_{ij}^{(t+1)} = 1$. The parameters of the distribution are initialized by $\theta_{ij}^{(0)} = K_i^{-1}$ if we have no prior knowledge. Moreover, we set the lower bound for $\theta_{ij}$ as $\theta_i^{\min} = (D(K_i - 1))^{-1}$ to leave open the possibility of generating all values (see Appendix B for details).

**Learning Rate Adaptation for Stochastic Natural Gradient Ascent:** The update formula (4) requires two hyper-parameters, namely the learning rate $\eta_\theta$ and the number $\lambda$ of Monte-Carlo samples. For Bernoulli distribution, Shirakawa et al. (2018) set $\lambda = 2$ and $\eta_\theta = D^{-1}$, the former of which is the minimum requirement to use a ranking-based utility transformation. As Bernoulli distribution is a special case of the categorical distribution with $K_i = 2$ for all $i$, the learning rate setting may be generalized as $\eta_\theta = \left( \sum_i^D (K_i - 1) \right)^{-1}$. However, as is not difficult to imagine, an adequate value for $\eta_\theta$ depends heavily on problem characteristic and stage of optimization. Nishida et al. (2018) proposed to adapt $\eta_\theta$ so as to keep the signal-to-noise ratio of successive parameter updates. It has been shown that the adaptive learning rate typically speeds up the convergence of the parameter vectors in the case of Bernoulli distributions. The adaptation mechanism can be applied to an arbitrary exponential family with expectation parameterization, which includes our categorical distribution. We employ this learning rate adaptation in our experiments. See Appendix C for details. From the preliminary experiment in the classification task, we confirmed that the learning rate adaptation improves the quality and stability of the architecture search. In the learning rate adaptation, the sample size $\lambda$ is fixed to two. From the viewpoint of stochastic approximation theory, the small learning rate has a similar effect of the large sample size. Therefore, even if the sample size equals two, we can optimize the distribution parameters properly with the appropriate small learning rate. In fact, we observed that the learning rate decreases by the learning rate adaptation mechanism.

**Overall Algorithm:** The overall algorithm of PDAS is displayed in Algorithm 1. We calculate the gradients of $W$ and $\theta$ using different mini-batches from a dataset. ENAS (Pham et al., 2018) and differentiable architecture search (DARTS) (Liu et al., 2018b), the methods for optimizing the structure parameters within a single training, use different datasets for the weight and structure parameter optimizations as well. Following these studies, we split the training dataset into halves

---

**Algorithm 1** PDAS with learning rate adaptation

---

**Require:** training data $\mathcal{D} = \{\mathcal{D}_W, \mathcal{D}_\theta\}$
1: Initialize the weight and distribution parameters as $W^{(0)}$ and $\theta^{(0)}$ and the parameters used in the learning rate adaptation
2: **for** $t = 0 \cdots T$ **do**
3:      Sample $M_1$ and $M_2$ independently from $p_{\theta^{(t)}}$
4:      Compute (2) using $M_1$, $M_2$, and a mini-batch from $\mathcal{D}_W$
5:      Update $W^{(t)}$ by a SGD method
6:      Update $\theta^{(t)}$ by (4) using $M_1$, $M_2$, and a mini-batch from $\mathcal{D}_\theta$ with updated $W$
7:      Update $\eta_\theta^{(t)}$ by the learning rate adaptation mechanism (See Appendix C for details)
8: **end for**
9: Get the structure parameters as $M^* = \mathrm{argmax}_M \, p_{\theta^{(T)}}(M)$
10: Re-train the weights using the fixed architecture represented by $M^*$

---

as $\mathcal{D} = \{\mathcal{D}_W, \mathcal{D}_\theta\}$. The gradients (2) and (3) are calculated using mini-batches from $\mathcal{D}_W$ and $\mathcal{D}_\theta$, respectively, and the parameter updates are performed alternately. As each dataset is sampled from the original one, the losses of mini-batch samples from both datasets approximate the original loss of all of the data if the dataset size is sufficiently large. Therefore, even if we use split datasets, we can view that the losses in the equations of (2) and (3) approximate the original loss.[1] Note that we do not need the back-propagation for calculating (3). Namely, the computational cost of the $\theta$ update is less than that of $W$.

After the optimization of $W$ and $\theta$, we can get the most likely structure parameters as $M^* = \mathrm{argmax}_M \, p_\theta(M)$, which is obtained trivially for the multivariate categorical distribution. Given the optimized structure parameters $M^*$, we *re-train* the neural network represented by $M^*$ with initialized weights. In the re-training stage, we can exclude the redundant weights (the weight parameters in the unused layer modules) and no longer need to update them. Re-training the obtained architecture is a commonly used technique (Brock et al., 2018; Liu et al., 2018b; Pham et al., 2018) to improve final performance. We have experimentally observed that the re-training of $W$ can improve the predictive performance.

## 3 RELATED WORK

The ordinary architecture search methods (Real et al., 2017; Suganuma et al., 2017; Zoph & Le, 2017) for deep neural networks repeat the following steps: the architecture generation, the weight training, and the architecture evaluation. Since the weight training of deep neural networks is time-consuming, the overall process requires a tremendous computational cost. Several techniques are introduced to reduce the training cost, such as inheriting the trained weights to the next candidate architectures (Real et al., 2017) and stopping the weight training based on the performance prediction (Baker et al., 2018). In contrast, our method is computationally more efficient than these approaches because it only needs the training twice (including the re-training).

The existing methods similar to our method are SMASH (Brock et al., 2018) and ENAS (Pham et al., 2018). SMASH randomly samples an architecture in memory-bank representation and determines its weights by a meta-network called HyperNet. Instead of training the weights in the generated network, the HyperNet is trained through back-propagation. Differently from our method, the probability distribution of network architectures does not change in SMASH, and it still needs the meta-network design. ENAS is a method based on the neural architecture search (NAS) (Zoph & Le, 2017). NAS defines a recurrent neural network, called the controller, that generates a sequence of categorical variables representing architecture for the main task and optimizes the controller using the policy gradient method in a hyper-parameter optimization manner. ENAS shares the weight parameters in all generated architectures and optimizes the weights and the controller parameters alternately.

---

[1]We can formulate the update rules with different datasets by starting from different original objectives for the weight and distribution parameters as done in ENAS.

Our method is similar to ENAS from the viewpoint of optimizing both of the weights and distribution parameters in a single training. The main difference between PDAS and ENAS is the probabilistic model of architectures, i.e., PDAS uses the categorical distribution, and ENAS uses the recurrent neural network. Since our method uses the categorical distribution, the modeling of architectures is intuitive and simple, and all we need is to design the categorical variables for representing architectures. As a result, we do not need to design the architecture of the controller neural network that is required for ENAS. In addition, the simple modeling of architecture makes it possible to derive the natural gradient in PDAS. As ENAS uses the LSTM network as the controller, it cannot derive the analytical natural gradient. Also, our method does not require to design the special operator for architecture search, such as crossover and mutation used in the evolutionary algorithms (Real et al., 2017). By injecting the learning adaptation mechanism (Nishida et al., 2018), we can further reduce the effort of the hyper-parameter tuning for PDAS. This property is convenient in practice. In the experiments, we always use the same hyper-parameter setting for PDAS. Another attractive property of PDAS is that the promising architecture is determined easily after the training described before, whereas it is difficult in ENAS.

## 4 EXPERIMENT AND RESULT

We apply the proposed method, the probabilistic model-based dynamic architecture search (PDAS), to the task of finding better architectures for image classification and inpainting. In the architecture search, two research directions exist: developing an efficient search method (Liu et al., 2018b; Pham et al., 2018) and designing a search space (Liu et al., 2018a; Zoph et al., 2018). Since PDAS is a search framework for neural network architectures, we concentrate on evaluating PDAS in terms of search efficiency: the quality of found architecture and the computational cost. Therefore, the experiment adopts the search spaces provided in the previous works, Pham et al. (2018) for classification and Suganuma et al. (2018) for inpainting. All experiments are run using a single NVIDIA GTX 1080Ti GPU, and PDAS is implemented using `PyTorch 0.4.1` (Paszke et al., 2017).

### 4.1 IMAGE CLASSIFICATION

**Dataset:** We use the CIFAR-10 dataset which consists of 50,000 and 10,000 RGB images of 32 $\times$ 32, for training and testing. All images are standardized in each channel by subtracting the mean and then dividing by the standard deviation. We adopt the standard data augmentation for each training mini-batch: padding 4 pixels on each side, followed by choosing randomly cropped 32 $\times$ 32 images and by performing random horizontal flips on the cropped images. We also apply the cutout (DeVries & Taylor, 2017) to the training data.

**Search Space:** The search space is based on the one in Pham et al. (2018) and the author's code[2], which consists of models obtained by connecting two motifs (called normal cell and reduction cell) repeatedly. An example of the overall model structure can be found in Appendix D. Each cell consisted of $B (= 5)$ nodes and receives the outputs of the previous two cells as inputs. Each node receives two inputs from previous nodes, applies an operation to each of the inputs, and adds them. Our search space includes 5 operations: identity, $3 \times 3$ and $5 \times 5$ separable convolutions (Chollet, 2017), and $3 \times 3$ average and max poolings. The separable convolutions are applied twice in the order of ReLU-Conv-BatchNorm. We select a node by 4 categorical variables representing 2 outputs of the previous nodes and 2 operations applied to them. Consequently, we treat $4B$-dimensional categorical variables for each cell. After deciding $B$ nodes, all unused outputs of the nodes are concatenated as the output of the cell. In the reduction cell, all operations applied to the inputs of the cell have a stride of 2. The number of the categorical variables is $D = 40$, and the dimension of $\theta$ becomes 180.

**Training Detail:** In the architecture search phase, we optimize $W$ and $\theta$ for 200 epochs with mini-batch size of 64. We stack 2 normal cells ($N = 2$) and set the number of channels at the first cell to 16. For the purpose of absorbing effect of the dynamic change in architecture, we fix affine parameters of batch normalizations. We use SGD with a momentum of 0.9 to optimize $W$. The learning rate changes from 0.025 to 0 followed the cosine schedule (Loshchilov & Hutter, 2017).

---

[2]https://github.com/melodyguan/enas

Table 1: Comparison with other architecture search methods on CIFAR-10. The notation "+c/o" indicates the cutout (DeVries & Taylor, 2017). The search cost indicates the GPU days for the architecture search, i.e., without the re-training cost. The result of the architecture randomly sampled from our search space is also listed as RANDOM.

| Method | Search Cost (GPU days) | Params (M) | Test Error (%) |
|---|---|---|---|
| NAS (Zoph & Le, 2017) | 16.8–22.4K | 7.1 | 4.47 |
| EAS (DenseNet) (Cai et al., 2018) | 20 | 10.7 | 3.44 |
| SMASHv2 (Brock et al., 2018) | 1.5 | 16.0 | 4.03 |
| NASNet-A + c/o (Zoph et al., 2018) | 2000 | 3.3 | 2.65 |
| NAONet + c/o (Luo et al., 2018) | 200 | 128 | 2.07 |
| NAONet-WS (Luo et al., 2018) | 0.4 | 3.7 | 3.53 |
| RENASNet + c/o (Chen et al., 2018) | 6.0 | 3.5 | 2.98 ($\pm$0.08) |
| DARTS first order + c/o (Liu et al., 2018b) | 1.5 | 2.9 | 2.94 |
| DARTS second order + c/o (Liu et al., 2018b) | 4 | 3.4 | 2.83 ($\pm$0.06) |
| ENAS + c/o (Pham et al., 2018) | 0.45 | 4.6 | 2.89 |
| RANDOM + c/o | − | 3.21 | 4.12 ($\pm$0.44) |
| PDAS + c/o | 0.24 | 3.20 | 2.98 ($\pm$0.12) |

We apply weight decay of $3 \times 10^{-4}$ and clip the norm of gradient at 5. In the re-training phase, we optimize $W$ for 600 epochs with mini-batch size of 80. We stack 6 normal cells ($N = 6$) and increase the number of channels at the first cell so that the model of the obtained architecture $M^*$ has nearly three million weight parameters. In contrast to the architecture search phase, we make all batch normalizations have learnable affine parameters because the architecture no longer changes. We apply the ScheduledDropPath (Zoph et al., 2018) dropping out each path between nodes, and the drop path rate linearly increases from 0 to 0.3 during the training. We also add the auxiliary classifier (Szegedy et al., 2016) with the weight of 0.4 that is connected from the second reduction cell. The total loss is a weighted sum of the losses of the auxiliary classifier and output layer. Other settings are the same as the architecture search phase. We conducted the experiment five times and report the average values.

**Result and Discussion:** Table 1 shows the comparison with other architecture search methods. Since the search space of the first three methods is different from the one used in PDAS, we should not directly compare the test errors to PDAS. At least, we can say that NAS and EAS require much more computational cost than PDAS because they repeat training the model many times to optimize network architecture. Although the search cost of SMASH is reasonable, PDAS is still faster than SMASH.

The other conventional methods adopt search spaces similar to PDAS.[3] Compared with these methods, our method is the fastest and shows a comparable error rate to ENAS, DARTS, and RENASNet. The architecture search of PDAS is realized by optimizing $\theta$ using (4), and its computational cost is small, whereas ENAS updates the controller recurrent network that generates the structure parameters. This might be one reason that PDAS is fast. DARTS models the architecture by the mixture of all possible operations and connections and optimizes the weights and the continuous structure parameters (mixture coefficients) by the gradient-based optimization. In the architecture search phase, DARTS requires to compute all possible operations and connections to calculate the gradient. In contrast, since PDAS computes the gradient using a few sampled structures, it is computationally more efficient than DARTS. The error rates of NASNet and NAONet outperform our method, but they have enormous search costs if they are implemented on a single GPU. We observe PDAS can take a good balance between the test error rate and search cost. The method denoted RANDOM uses the architecture randomly sampled from our search space. The result shows that PDAS can find the better architecture in a reasonable computational time by optimizing the structure parameters in the single training.

---

[3]The setting, e.g., types of operations, channel sizes, and the number of normal cells $N$ differs among the methods.

We observed that the value of $\theta$ converges to a certain category, and the average value of $\max_j \theta_{ij}$, implying the convergence of the parameters of the distribution approaches 0.9 at 50th epoch. The architecture of the best model obtained by PDAS appears in Appendix E. We note that PDAS has fewer hyper-parameters and can be used by only providing the categorical variables for representing the architecture, whereas, other methods still leave the controller network design or strategy parameter tuning. As our algorithm is based on the stochastic natural gradient method, we can easily add the improving techniques, such as the learning rate adaptation used in PDAS.

## 4.2 INPAINTING

The inpainting task is one of the image restoration tasks, restoring a clean image from a damaged image with large missing regions (e.g., masks). Suganuma et al. (2018) have shown the potential of the architecture search by the evolutionary algorithm for image restoration tasks including inpainting. In this section, we apply the PDAS to the problem of architecture search for inpainting and evaluate its performance. We refer the experimental setting employed in Suganuma et al. (2018).

**Dataset and Evaluation Measure:** We use three benchmark datasets: the CelebFaces Attributes Dataset (CelebA) (Liu et al., 2015), the Stanford Cars Dataset (Cars) (Krause et al., 2013), and the Street View House Numbers (SVHN) (Netzer et al., 2011). The CelebA is a large-scale human face image dataset that contains 202,599 RGB images. We select 101,000 and 2,000 images for training and test, respectively, in the same way as Suganuma et al. (2018). All images were cropped to properly contain the entire face by using the provided bounding boxes and resized to $64 \times 64$ pixels. The Cars is a middle-scale cars image dataset that contains 16,185 RGB images, and it consists of 8,144 and 8,041 images for training and testing, respectively. Similar to the CelebA, all images were cropped by using the provided bounding boxes and resized to $64 \times 64$ pixels. The SVHN is a large-scale house-number image dataset that contains 99,289 RGB images without extra training data, and it consists of 73,257 and 26,032 images for training and testing, respectively. The images of SVHN were resized to $64 \times 64$ pixels. All images are normalized by dividing by 255, and we perform data augmentation of random horizontal flipping on the training images.

We test three different masks based on Suganuma et al. (2018); Yeh et al. (2017); a central square block mask (Center); a random pixel mask, as 80% of all the pixels were randomly masked (Pixel); and a half image mask, as a randomly selected vertical or horizontal half of the image (Half). The mask was randomly generated for each training mini-batch and each test image.

Following Suganuma et al. (2018), we use two standard evaluation measures: the peak-signal to noise ratio (PSNR) and the structural similarity index (SSIM) (Wang et al., 2004) to evaluate the restored images. Higher values of these measures indicate a better image restoration.

**Search Space:** The search space we use is based on Suganuma et al. (2018). We employ the CAE, which is similar to the RED-Net (Mao et al., 2016), as a base architecture. The RED-Net consists of a chain of convolution layers and symmetric deconvolution layers as the encoder and decoder parts, respectively. The encoder and decoder parts perform the same counts of downsampling and upsampling with a stride of 2, and a skip connection between the convolutional layer and the mirrored deconvolution layer can exist. For simplicity, each layer employs either a skip connection or a downsampling, and the decoder part is employed in the same manner. In the skip connected deconvolution layer, the input feature maps from the encoder part are added to the output of deconvolution operation followed by ReLU. In the other layers, the ReLU activation is performed after the convolution and deconvolution operations. We prepare six types of layers: the combination of the kernel sizes $\{1 \times 1, 3 \times 3, 5 \times 5\}$ and the existence of the skip connection. The layers with different settings do not share weight parameters.

Since we consider the symmetric CAE, it is enough to represent the encoder part; we only need to determine the encoder part of the CAE by the categorical variables, and then the decoder part is automatically decided according to the encoder part. We consider $N_c$ hidden layers and the output layer. We encode the type, channel size, and connections of each hidden layer. The kernel size and stride of the output deconvolution layer are fixed with $3 \times 3$ and 1, respectively; however, the connection is determined by a categorical variable. The numbers of categories for the hidden layer type and the output channel size are 6 and 3, respectively. We select the output channel size of each layer from $\{64, 128, 256\}$. To ensure the feed-forward architecture and to control the network depth,

Table 2: Comparison of PSNR and SSIM in the inpainting tasks. CE and SII indicate the context encoder (Pathak et al., 2016) and the semantic image inpainting (Yeh et al., 2017), which are CNN-based methods. E-CAE means the model obtained by the architecture search method using the evolutionary algorithm (Suganuma et al., 2018). BASE is the same depth of the best architecture obtained by E-CAE but having the 64 channels and $3 \times 3$ filters in each layer with a skip connection. Note that CE and SII used human-designed architecture, while E-CAE and PDAS (Ours) are the models obtained by the architecture search. The values of CE, SII, BASE, and E-CAE are referenced from Suganuma et al. (2018).

| Dataset | Mask | PSNR [dB] / SSIM | | | | |
| | | CE | SII | BASE | E-CAE | PDAS |
|---|---|---|---|---|---|---|
| CelebA | Center | 28.5 / 0.912 | 19.4 / 0.907 | 27.1 / 0.883 | 29.9 / 0.934 | 28.5 / 0.896 |
| | Pixel | 22.9 / 0.730 | 22.8 / 0.710 | 27.5 / 0.836 | 27.8 / 0.887 | 26.9 / 0.871 |
| | Half | 19.9 / 0.747 | 13.7 / 0.582 | 11.8 / 0.604 | 21.1 / 0.771 | 20.3 / 0.771 |
| Cars | Center | 19.6 / 0.767 | 13.5 / 0.721 | 19.5 / 0.767 | 20.9 / 0.846 | 19.8 / 0.751 |
| | Pixel | 15.6 / 0.408 | 18.9 / 0.412 | 19.2 / 0.679 | 19.5 / 0.738 | 18.9 / 0.677 |
| | Half | 14.8 / 0.576 | 11.1 / 0.525 | 11.6 / 0.541 | 16.2 / 0.610 | 15.7 / 0.596 |
| SVHN | Center | 16.4 / 0.791 | 19.0 / 0.825 | 29.9 / 0.895 | 33.3 / 0.953 | 32.4 / 0.945 |
| | Pixel | 30.5 / 0.888 | 33.0 / 0.786 | 40.1 / 0.899 | 40.4 / 0.969 | 42.6 / 0.986 |
| | Half | 21.6 / 0.756 | 14.6 / 0.702 | 12.9 / 0.617 | 24.8 / 0.848 | 24.2 / 0.854 |

the connection of $i$-th layer is only allowed to be connected from $(i - 1)$ to $\max(0, i - b)$-th layers, where $b$ ($b > 0$) is called the level-back parameter. Namely, the categorical variable representing the connection of $i$-th layer has $\max(i, i - b)$ categories. Obviously, the first hidden layer always connects with the input, and we can ignore this part. With this representation, it has the possibility to exist the *inactive* layers which do not connect to the output layer. Therefore, this model can represent variable length architectures by the fixed length categorical variables. The illustration of an example of the decoded CAE and corresponding categorical variables is displayed in Appendix D. We choose $N_c = 20$ and the level-back parameter $b = 5$. With this setting, the number of categorical variables is $D = 60$, and the dimension of $\theta$ becomes 274.

**Training Detail:** We use the mean squared error (MSE) as the loss $\mathcal{L}$ and the mini-batch size of $\bar{N} = 16$. The weight parameter $W$ is optimized by SGD with a momentum of 0.9. The learning rate changes from 0.025 to 0 followed the cosine schedule (Loshchilov & Hutter, 2017) without restart mechanism. We apply gradient clipping with the norm of 5 to prevent too long gradient step. The maximum numbers of iterations are $50K$ and $500K$ in the architecture search and re-training phases, respectively. The number of iterations for re-training is the same setting as in Suganuma et al. (2018).

**Result and Discussion:** Table 2 shows the comparison of PSNR and SSIM. The human-designed CNN-based methods (CE and SII) and the architecture search method (E-CAE) proposed in Suganuma et al. (2018) are listed. Also, the architecture having the same depth of the best architecture obtained by E-CAE but having the 64 channels and $3 \times 3$ filters in each layer with a skip connection (BASE). The performance of PDAS is better than CE, SII, and BASE on almost dataset and mask types. Furthermore, PDAS shows good performance for all settings, as is the case with E-CAE. Compared with E-CAE, the performance of PDAS is a bit worse; however, PDAS outperforms E-CAE for SVHN with the random pixel mask, regarding both of PSNR and SSIM. PDAS shows the intermediate performance between BASE and E-CAE. Suganuma et al. (2018) reported that E-CAE spent approximately 12 GPU days (3 days with 4 GPUs) for the architecture search and re-training. However, the average computational time of PDAS is approximately 7 hours for the architecture search and 13 hours for the re-training (totally 0.83 GPU days). Since the performance and search cost have a trade-off relation in general, PDAS is a reasonable and promising architecture search method. In the architecture search, one of the important points for the final performance is the search space design, e.g., the design of modules and the encoding scheme of the architecture. Practical users can spend the spare time of PDAS to try other search spaces.

## 5 CONCLUSION

We proposed an efficient architecture search method called PDAS that optimizes the parameters of the categorical distribution based on the stochastic natural gradient method during weight training. Our probabilistic modeling of the architecture is straightforward, and the derived algorithm has fewer hyper-parameters and can incorporate the learning rate adaptation mechanism. The experimental results for the image classification and inpainting tasks have shown that PDAS is fast and achieves comparable performance to the existing methods. In the future, we will apply PDAS to other neural network architecture searches (e.g., the recurrent networks) and large-scale datasets (e.g., ImageNet). Regarding the extension of PDAS, the distribution of continuous variables can be introduced to PDAS; then we will be able to optimize the architecture represented by mixed variables (e.g., categorical and continuous variables).

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

## A   DERIVATION OF THE NATURAL GRADIENT

We derive the natural gradient of the log-likelihood associated with our categorical distributions. We first reduce the number of parameters by $D$, number of variables, by forcing the parameter value for the last category of each variable, namely $\theta_{iK_i} = 1 - \sum_{k=1}^{K_i-1} \theta_{ik}$. It does not decrease the degrees of freedom of the family of distributions. We let the reduced parameter and corresponding random variable denoted by $\bar\theta$ and $\bar M$, respectively. The probability mass reads

$$p_{\bar\theta}(\bar M) = \prod_{i=1}^{D} \left( \prod_{j=1}^{K_i-1} (\theta_{ij})^{m_{ij}} \right) \left( 1 - \sum_{j=1}^{K_i-1} \theta_{ij} \right)^{\left(1-\sum_{j=1}^{K_i-1} m_{ij}\right)} . \tag{5}$$

We continue to represent the deleted variables and parameters as $m_{iK_i}$ and $\theta_{iK_i}$ for the simple notation. The element of derivative of log-likelihood is given by $\nabla_{\theta_{ij}} \ln p_{\bar\theta}(\bar M) = \frac{m_{ij}}{\theta_{ij}} - \frac{m_{iK_i}}{\theta_{iK_i}}$.

The Fisher information matrix, $\mathbb{E}[\nabla_{\bar\theta} \ln p_{\bar\theta}(M) \nabla_{\bar\theta} \ln p_{\bar\theta}(M)^{\mathrm T}]$, is a block-diagonal matrix whose $i$-th diagonal block $F_i(\bar\theta)$ ($i = 1, \ldots, D$) is of form

$$F_i(\bar\theta) = \mathrm{diag}(\bar\theta_i)^{-1} + \frac{1}{\theta_{iK_i}} \mathbf{1}\mathbf{1}^{\mathrm T} , \tag{6}$$

where $\mathrm{diag}(\bar\theta_i)$ is the diagonal matrix whose diagonal elements are the corresponding elements of $\bar\theta_i = (\theta_{i1}, \ldots, \theta_{iK_i-1})$ and $\mathbf 1$ is the vector consisting of $1$ for all entries. Then, it is easy to see that its inverse is again block-diagonal, and each block reads (thanks to the Sherman-Morrison formula)

$$F_i^{-1}(\bar\theta) = \mathrm{diag}(\bar\theta_i) - \bar\theta_i \bar\theta_i^{\mathrm T} . \tag{7}$$

The natural gradient, the product of the inverse Fisher information matrix and the vanilla gradient, can be written as

$$\tilde\nabla_{\bar\theta} \ln p_{\bar\theta}(M) = F^{-1}(\bar\theta) \nabla_{\bar\theta} \ln p_{\bar\theta}(M) = \bar M - \bar\theta . \tag{8}$$

With the learning rate $\eta_\theta$, the $\bar\theta$ update rule is written as

$$\bar\theta^{(t+1)} = \bar\theta^{(t)} + \frac{\eta_\theta}{\lambda} \sum_{n=1}^{\lambda} u_n(\bar M_n - \bar\theta^{(t)}) . \tag{9}$$

The update of $\theta_{iK_i}^{(t+1)} = 1 - \sum_{k=1}^{K_i-1} \theta_{ik}^{(t+1)}$ reads

$$\theta_{iK_i}^{(t+1)} = \theta_{iK_i}^{(t)} + \frac{\eta_\theta}{\lambda} \sum_{n=1}^{\lambda} u_n((m_{iK_i})_n - \theta_{iK_i}^{(t)}) . \tag{10}$$

As a consequence, the $\theta$ update rule is reduced to (4).

## B   RESTRICTION FOR THE RANGE OF $\theta$

To guarantee a small yet positive probability for all combinations of categorical variables, the parameters are projected into a subset of the domain of the parameters, namely, $\left[ \theta_i^{\min} := \frac{1}{D(K_i-1)}, 1 - \frac{1}{D} \right]$ for each $i = 1, \ldots, D$, where $D$ is the number of categorical variables and $K_i > 1$ is the number of categories for $i$-th variable. To realize this, we apply the following steps after $\theta$ update by (4):

$$\theta_{ij} \leftarrow \max\{\theta_{ij}, \theta_i^{\min}\} \text{ for all } i \text{ and } j, \text{ then} \tag{11}$$

$$\theta_{ij} \leftarrow \theta_{ij} + \frac{1 - \sum_{k=1}^{K_i} \theta_{ik}}{\sum_{k=1}^{K_i} \left( \theta_{ik} - \theta_i^{\min} \right)} \left( \theta_{ij} - \theta_i^{\min} \right) . \tag{12}$$

The first line guarantees $\theta_{ij} \geq \theta_i^{\min}$. The second line ensures $\sum_{j=1}^{K_i} \theta_{ij} = 1$, while keeping $\theta_{ij} \geq \theta_i^{\min}$.

## C  LEARNING RATE ADAPTATION

The learning rate adaptation proposed by Nishida et al. (2018) is adopted to achieve the parameter-free algorithm and improve the search efficiency. Let $s$ be the accumulation of the parameter update and $\gamma_s$ be its normalization factor, initialized as $s^{(0)} = \mathbf{0}$ and $\gamma_s^{(0)} = 0$, respectively. We denote the estimated natural gradient in (4) as $\tilde{\nabla}_\lambda^{(t)} = \frac{1}{\lambda} \sum_{n=1}^{\lambda} u_n(\bar{M}_n - \bar{\theta}^{(t)})$. Note that $M$ and $\theta$ are over-parametrized by the one degree of freedom as mentioned in the natural gradient derivation. We hence consider $\bar{M}$ and $\bar{\theta}$ as we considered above. Then, they are updated as

$$s^{(t+1)} = (1 - \eta_\theta^{(t)})s^{(t)} + \sqrt{\eta_\theta^{(t)}(2 - \eta_\theta^{(t)})}\frac{F(\theta^{(t)})^{\frac{1}{2}}\tilde{\nabla}_\lambda^{(t)}}{\text{Tr}(F(\theta^{(t)})\text{Cov}[\tilde{\nabla}_\lambda^{(t)}])^{\frac{1}{2}}}$$

$$\gamma_s^{(t+1)} = (1 - \eta_\theta^{(t)})^2\gamma_s^{(t)} + \eta_\theta^{(t)}(2 - \eta_\theta^{(t)}) \ .$$

The estimated natural gradient is scaled w.r.t. the Fisher information matrix and its estimation co-variance. As we can not compute the covaraince matrix analytically, we approximate it by the expectation under the assumption that $u_n$ and $M_n$ are uncorrelated, leading to $\text{Tr}(F(\theta^{(t)})\text{Cov}[\tilde{\nabla}_\lambda^{(t)}]) = 2^{-1}\sum_{i=1}^{D}(K_i - 1)$, where we used the fact $\lambda = 2$, $u_1 = 1$ and $u_2 = -1$. See Nishida et al. (2018) for its rationale. The learning rate is updated based on the length of $s^{(t+1)}$, namely,

$$\eta_\theta^{(t+1)} = \eta_{\min} \vee \eta_\theta^{(t)} \exp(\eta_\theta^{(t)}(\|s^{(t+1)}\|^2/\alpha - \gamma_s^{(t+1)})) \wedge \eta_{\max} \ , \tag{13}$$

where $\alpha = 1.5$ is the threshold parameter, $\eta_{\min}$ and $\eta_{\max}$ are the minimum and maximum learning rate, which are $\eta_{\min} = 0$ and $\eta_{\max} = (\sum_{i=1}^{D}(K_i - 1))^{-1/2}$, respectively. The learning rate is initialized as $\eta_\theta^{(0)} = \eta_{\max}$.

The above procedure requires to compute the square root of $F(\theta^{(t)})$, which is feasible since it is positive definite. As the Fisher information matrix is a block-diagonal, and each block is of size $K_i - 1$, a naive computation of $F(\theta^{(t)})^{\frac{1}{2}}$ requires $O(\sum_{i=1}^{D}(K_i - 1)^3)$. This is usually not expensive as $D \gg K_i$. An alternative way that we employ in this paper is to replace $F(\theta^{(t)})^{\frac{1}{2}}$ with a tractable factorization $A$ with $F(\theta^{(t)}) = AA^{\text{T}}$. Our choice of $A$ is the block-diagonal matrix whose $i$-th block is square, of size $K_i - 1$, and

$$A_i = \text{diag}(\bar{\theta}_i)^{-\frac{1}{2}} + \frac{1}{(\sqrt{\theta_{iK_i}} + \theta_{iK_i})}\mathbf{1}\sqrt{\bar{\theta}_i}^{\text{T}} \ , \tag{14}$$

where $\sqrt{\bar{\theta}_i}$ is a vector whose $j$-th element is the square root of $\bar{\theta}_{ij}$. Then, the product of $A$ and a vector can be computed in $O(\sum_{i=1}^{D}(K_i - 1))$. In our preliminary study we did not obverse any significant performance difference by this approximation.

## D  OVERALL MODEL STRUCTURE FOR CLASSIFICATION AND INPAINTING TASKS



Figure 1: Overall model structure for the classification task in Section 4.1. We optimize the architecture of the normal and reduction cells by PDAS. In the re-training phase, we construct the CNN using the optimized cell architecture with an increased number of cells $N$.

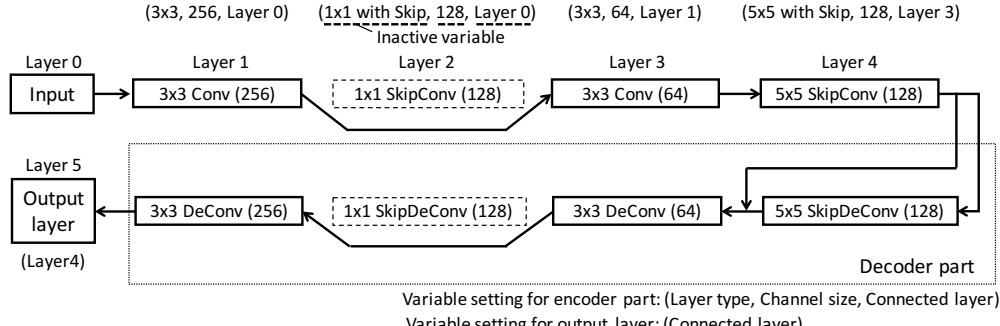

Figure 2: A conceptual example of the decoded symmetric CAE architecture and the corresponding categorical variables. The decoder part is automatically decided from the encoder structure as a symmetric manner.

# E SUPPLEMENTARY EXPERIMENTAL RESULTS

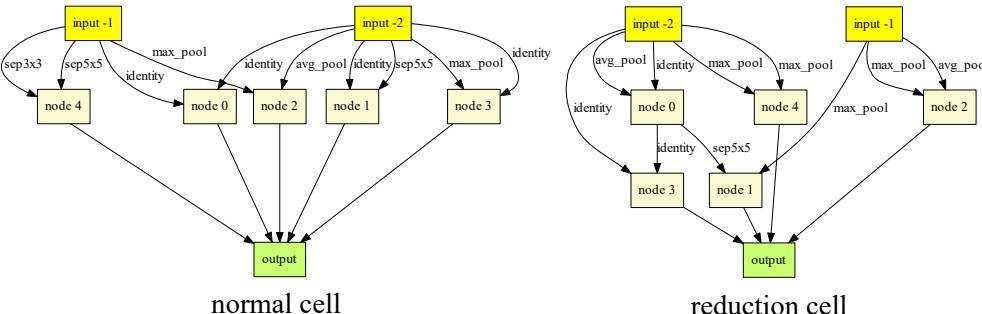

Figure 3: The best cell structures discovered by PDAS in the classification task.

