# OpenReview forum: "Probabilistic Model-Based Dynamic Architecture Search"
_ICLR.cc/2019/Conference_

### Official Review · AnonReviewer3 · 2018-11-02
**Lacking novelty, but cool results**

**Rating:** 5
**Confidence:** 4

**Review:**

This paper presents a joint optimization approach for the continuous weights and categorical structures of neural networks. The idea is the standard stochastic relaxation of introducing a parametrised distribution over the categorical parameters and marginalising it. The method then follows by alternating gradient descent on the weights and the parameters of the categorical distribution.

This exact approach was proposed in https://arxiv.org/abs/1801.07650 by Shirakawa et al. The only innovation in this work is that it uses categorical distributions with more than two values. This is a minor innovation.

The experiments are however interesting as the paper compares to the latest hyper-parameters optimization strategies for neural nets on simple tasks (eg CIFAR10) and gets comparable results. However, given that this is the biggest contribution of the paper, it would have been nice to see results in more complex tasks, eg imagenet or translation.

I very much enjoyed the simplicity of the approach, but the question of innovation is making wonder whether this paper makes the ICLR bar of acceptance. The paper is also hard to read because of many English typos.

---

> ### Author Response · Authors · 2018-11-26
> **Reply to reviewer 3**
>
> We thank reviewers for reviewing this paper and appreciate their pointing out important aspects. We reply to the reviewer’s comments.
>
> Our method is an extension of Shirakawa et al. (2018), and the resulting simple algorithm has fewer hyper-parameters than other architecture search methods but can reach state-of-the-art performance with low computational cost. We derived the algorithm for categorical distribution and made it possible to apply the framework to architecture search spaces represented by categorical variables. To the best of our knowledge, the natural gradient of categorical distribution has not been introduced in the context of stochastic natural gradient methods.
>
> We believe that the simplicity of architecture search methods is an important aspect. It is helpful to reduce the effort of the hyper-parameter tuning of the architecture search method itself. The experimental result implies that the architecture search method does not have to be complicated.
>
> Our experiments consist of CIFAR-10 classification and inpainting. The CIFAR-10 classification is a well-studied and relatively simple task, but inpainting is considered a complicated task. As the reviewer pointed out, applying PDAS to the ImageNet dataset or translation task is an important direction. Regarding the ImageNet dataset, as the work of Liu et al. (2018) showed, the architecture obtained in the CIFAR-10 can be transferred to the ImageNet dataset and works well. Since our architecture search space is almost the same as in Liu et al. (2018), the discovered architecture by PDAS can be transferred to ImageNet.
>
> Hanxiao Liu, Karen Simonyan, and Yiming Yang, “DARTS: Differentiable Architecture Search,” arXiv preprint:1806.09055 (2018)
>
>
> Also, we did our best to fix the English typos in the revised paper to improve the readability.

---

> > ### Comment · AnonReviewer3 · 2018-11-29
> > **Good work but it could be improved**
> >
> > I'd like to first thank the authors for their reply. They have tried conscientiously to improve the paper.
> >
> > However, in its current form, I believe the paper still has two shortcomings, namely the similarity to the work of Shirakawa et al (2018) and its comparison to ENAS. I think with a bit of thinking, you may find that there are tuning problems better fitted to PDAS than ENAS (eg entire architecture instead of a recurrent module). I strongly encourage you to pursue this. It is good work, but if improved, it will be more convincing and have far more impact.  I strongly encourage the authors to continue working on it and resubmit soon.

---

### Official Review · AnonReviewer2 · 2018-11-04
**Proposes an architecture search technique, easy to read. Not confident about the baselines and how this is compared to the literature.**

**Rating:** 6
**Confidence:** 4

**Review:**

This paper proposes an architecture search technique in which the hyperparameters are modeled as categorical distribution and learned jointly with the NN. The paper is written well. I am not an expert of the literature in this domain so will not be able to judge the paper regarding where it is located in the related work field.

Pros:
-This is a very important line of research direction that aims to make DNNs practical, easy to deploy and cost-effective for production pipelines.
-The categorical distribution for hyperparameters makes sense, and the derivation of the joint training seems original idea. I liked the fact that you need to train the NN just twice (the second one only to fine tune with optimized parameters)
-Two very different problems (inpainting/encoding-decoding + CNN/classification) have been demonstrated.
-Existing experiments have been explained with enough detail except for minor points.

Cons:
-I speculate that there is a trade-off between the number of different parameters and whether one training is good enough to learn the architecture distribution. i.e., When you have huge networks and many parameters, how well this method works? I think the authors could provide some experimental study suggesting their users what a good use case of this algorithm is compared to other techniques in the literature. In what type of network and complexity this search method works better than others?
-E-CAE for in-painting seems to be working significantly better than the proposed technique. Regarding results, I was expecting more insights into why this is the case. As above, at what type of a problem one should pick which algorithm? If the 7hours vs. 3days GPU difference negligible for a client, should one pick E-CAE?
-In theory, there has been shown lambda samples (equation 2 and 3). However, the algorithm seems to be using just 2? If I didn't miss, this is not discussed thoroughly. I speculate that this parameter is essential as the categorical distribution gets a bigger search space. Also the reliability of the model and final performance, how does it change concerning this parameter?

---

> ### Author Response · Authors · 2018-11-26
> **Reply to reviewer 2**
>
> We thank reviewers for reviewing this paper and appreciate their pointing out important aspects. We reply to the reviewer’s comments.
>
>
> (1) Trade-off between the number of distribution parameters and performance
> In general, we need a large number of iterations when we optimize a large number of distribution parameters. The previous study (Shirakawa et al. 2018) shows that the method could solve the problem of 3,000 bits in 1M iterations. We note that the number of bits corresponds to the total dimension of the one-hot vectors in our categorical distribution case. Considering the modern deep learning setting, the number of iterations for network training is up to about 1M. In summary, we can propose the guideline that PDAS should be applied to the problem with up to 3,000 dimensions.
>
>
> (2) Comparison with E-CAE in inpainting task
> As the reviewer pointed out, the performance of E-CAE is superior to our method, while the running time (GPU time) of E-CAE is about 14 times slower than PDAS. We note that PDAS can be parallelized for the sample size and mini-batch size. It means that PDAS users can try another setting and tune several hyper-parameters by using this spare time. One of the important points for the final performance is the search space design, e.g., the design of modules and the encoding scheme of the architecture. Since we propose the architecture search method in this paper, we applied the proposed method to the same search space used in the previous studies. However, practical users can spend the spare time to try other search spaces.
>
> We have added the explanation about this point to Section 4.
>
>
> (3) Sample size
> We used just two samples to estimate the gradients. From the viewpoint of stochastic approximation theory, the small learning rate has a similar effect of the large sample size　(Nishida et al. 2018). Therefore, even if the sample size equals two, we can optimize the distribution parameters properly with the appropriate small learning rate. In fact, we have observed that the learning rate decreases by the learning rate adaptation mechanism. In the CIFAR-10 case, the learning rate starts from 0.0845 and decreases to about 0.003. In the work of Nishida et al. (2018), the sample size adaptation mechanism for a fixed learning rate is also introduced. One possible future direction is to investigate the effect of the sample size adaptation instead of the learning rate adaptation.
>
> We have added the explanation about this point to Section 2.

---

### Official Review · AnonReviewer1 · 2018-11-07
**Simple and effective method with limited novelty**

**Rating:** 5
**Confidence:** 4

**Review:**

The authors propose to formulate the neural network architecture as a collection of multivariate categorical distributions. They further derive sample-based gradient estimators for both the stochastic architecture and the deterministic parameters, which leads to a simple alternating algorithm for architecture search.

Pros:
+ Intuitions and formulations are easy to comprehend.
+ Simpler to implement than most prior methods.
+ Appealing results (on CIFAR-10) as compared to the state-of-the-art.

Cons:
- Limited technical novelty. The approach is a straightforward extension of Shirakawa et al. 2018. The main algorithm is essentially the same except minor differences in gradient derivations.
- Lack of theoretical justifications. It seems all the derivations at the beginning of Section 2 assume the architecture is optimized wrt the training set. However, the authors ended up splitting the dataset into two parts in the experiments and optimize the architecture wrt a separate validation set instead. This would invalidate all the previous derivations.
- The method is a degenerated version of ENAS. A closer look at eq (2) and (3) suggests the resulting iterative algorithm is almost the same as that in ENAS, where the weights are optimized using GD wrt the training set and the architecture is optimized using the log-derivative trick wrt the validation set. The only distinction are (i) using a degenerated controller/policy formulated as categorical distributions (ii) using the validation loss instead of the validation accuracy as the reward (according to eq. (3)). This is also empirically reflected in Table 1, which shows the proposed PDAS is similar to ENAS both in terms of efficiency and performance. The mathematical resemblance with ENAS is not necessarily bad, but the authors need to make it more explicit in the paper.

Minor issues:
* I'm not sure whether it's a good practice to report the "best" test error among multiple runs in Table 1.
* The method is not really "parameterless" as claimed in the introduction. For example, a suitable learning rate adaptation rule can be task-specific thus requires manual tuning/design. The method also consists of some additional hyperparameters like the \lambda in the utility transform.

---

> ### Author Response · Authors · 2018-11-26
> **Reply to reviewer 1**
>
> We thank reviewers for reviewing this paper and appreciate their pointing out important aspects. We reply to the reviewer’s comments.
>
> First of all, we believe that it is worthwhile that the simple proposed method can achieve competitive performance with complicated architecture search methods. This implies that the previous methods are overcomplicated and simple probabilistic modeling is sufficient for architecture search. This aspect helps us to reduce the effort in tuning the hyper-parameters of the architecture search method itself. Also, the proposed method is the fastest among the existing architecture methods.
>
>
> (1) Novelty and contribution
> As the reviewer pointed out, this paper is an extension of Shirakawa et al. (2018). The previous work only derived the algorithm for Bernoulli distribution and only applied it to simple tasks, e.g., layer selection and connection pruning. The contribution of this paper is as follows:
>
> (i) We derived the algorithm for categorical distribution and made it possible to apply the framework to architecture search spaces represented by categorical variables. To the best of our knowledge, the natural gradient of categorical distribution has not been introduced in the context of stochastic natural gradient methods.
>
> (ii) We showed PDAS, which has fewer hyper-parameters than ENAS, is fast and can reach state-of-the-art performance. The intrinsic hyper-parameters of PDAS are the sample size and the learning rate, but the learning rate can be adaptive.
>
> We have added the explanation about the novelty and contribution to Section 1.
>
>
> (2) Theoretical justifications
> We split the dataset into two datasets with the same number of items for updating the weight and distribution parameters. As each dataset is sampled from the original one, the losses of mini-batch samples from both datasets approximate the original loss of all of the data if the dataset size is sufficiently large. Therefore, even if we use split datasets, we can view that the losses in the equations of (2) and (3) approximate the original loss. Of course, we can formulate the update rules with different datasets by starting from different original objectives for the weight and distribution parameters as done in ENAS. We have added the sentence about this point to Section 2.
>
>
> (3) Relation to ENAS
> As the reviewer pointed out, the main difference between PDAS and ENAS is the probabilistic model of architectures, i.e., PDAS uses the categorical distribution, and ENAS uses the LSTM network. We make this point clear by adding the sentence to Section 3.
>
> As the search space of CIFAR-10 is the same in ENAS and PDAS, we assume that both methods can find quasi-optimum architectures. We would like to emphasize that PDAS can provide competitive performance with lower computational cost compared to ENAS though it is simple. The simple modeling of architecture makes it possible to derive the natural gradient in PDAS. As ENAS uses the LSTM network as the controller, it cannot derive the analytical natural gradient. The natural gradient method can update the model parameters to the steepest direction with respect to KL-divergence and has a big advantage in the optimization of a probabilistic model.
>
> The intrinsic hyper-parameters of PDAS are the sample size and the learning rate. In addition, the learning rate can be adaptive by the method in Nishida et al. (2018). Meanwhile, ENAS has more hyper-parameters, e.g., the sample size, the learning rate, the regularization coefficient, the number of units in LSTM, and the architecture design of the controller. Moreover, ENAS tuned such hyper-parameters depending on the tasks, e.g., they used the regularization coefficients of 0.0001 for PTB and 0.1 for CIFAR-10. However, our method did not change the hyper-parameters for the image classification and inpainting tasks. Our method has such attractive properties because it is designed based on the stochastic natural gradient method, which is theoretically well studied, for example, in Akimoto and Ollivier (2013). We would like to emphasize that our method is handier than ENAS in practice because PDAS showed the decent performance without problem-specific hyper-parameter tuning as ENAS does.
>
> Y. Akimoto and Y. Ollivier, “Objective Improvement in Information-Geometric Optimization,” Foundations of Genetic Algorithms XII (FOGA XII) (2013).
>
> We have added the sentence about this point to Section 3.
>
>
> (4) Minor issues
> As the reviewer pointed out, it is meaningless to report the "best" test error. We have removed it from Table 1.
>
> Also, the proposed method is not really “parameterless.” Specifically, our method is parameter-adaptive or pseudo parameterless. We would like to say that our method can achieve decent performance with low computational cost without special hyper-parameter tuning. We have modified the explanation about this point in Section 1 and Section 5.

---

### Meta-Review · Area_Chair1 · 2018-12-13
**insufficient novelty**

**Confidence:** 5
**Recommendation:** Reject

**Metareview:**

The paper presents an architecture search method which jointly optimises the architecture and its weights. As noted by reviewers, the method is very close to Shirakawa et al., with the main innovation being the use of categorical distributions to model the architecture. This is a minor innovation, and while the results are promising, they are not strong enough to justify acceptance based on the results alone.